:ₒ: PLOS | ONE

# A new method of recording from the giant fiber of *Drosophila melanogaster* shows that the strength of its auditory inputs remains constant with age

Jonathan M. Blagburn[ORCID]*

Institute of Neurobiology, University of Puerto Rico Medical Sciences Campus, San Juan, PR, United States of America

* jonathan.blagburn@upr.edu

**Data Availability Statement:** All relevant data are within the manuscript and its Supporting Information files.

## Abstract

There have been relatively few studies of how central synapses age in adult *Drosophila melanogaster*. In this study we investigate the aging of the synaptic inputs to the Giant Fiber (GF) from auditory Johnston's Organ neurons (JONs). In previously published experiments an indirect assay of this synaptic connection was used; here we describe a new, more direct assay, which allows reliable detection of the GF action potential in the neck connective, and long term recording of its responses to sound. Genetic poisoning using diphtheria toxin expressed in the GF with *R68A06-GAL4* was used to confirm that this signal indeed arose from the GF and not from other descending neurons. As before, the sound-evoked action potentials (SEPs) in the antennal nerve were recorded via an electrode inserted at the base of the antenna. It was noted that an action potential in the GF elicited an antennal twitch, which in turn evoked a mechanosensory response from the JONs in the absence of sound. We then used these extracellular recording techniques in males and female of different ages to quantify the response of the JONs to a brief sound impulse, and also to measure the strength of the connection between the JONs and the GF. At no age was there any significant difference between males and females, for any of the parameters measured. The sensitivity of the JONs to a sound impulse approximately doubled between 1 d and 10 d after eclosion, which corresponds to the period when most mating is taking place. Subsequently JON sensitivity decreased with age, being approximately half as sensitive at 20 d and one-third as sensitive at 50 d, as compared to 10 d. However, the strength of the connection between the auditory input and the GF itself remained unchanged with age, although it did show some variability that could mask any small changes.

## Introduction

Changes in synaptic connectivity that occur with normal aging, rather than neurological disease, have been a focus of research for decades [1–9]. Even in "simple" model organisms, such

**Funding:** JMB is supported by National Institutes of Health, National Institute of General Medical Sciences grant SC3GM121190. The Institute Drosophila resource center is supported by National Institute on Minority Health and Health Disparities G12 MD007600 (Research Centers for Minority Institutions). The funders had no role in study design, data collection and analysis, decision to publish, or preparation of the manuscript"

**Competing interests:** The authors have declared that no competing interests exist.

as *Drosophila melanogaster*, changes in neuromuscular synaptic structure and function have been seen with aging [10–12]. Central synaptic circuits in the fly have been less well studied, although the synaptic changes underlying age-related memory deficits have been investigated in detail [13,14]. Recently, however, it was shown that there is an age-related decline in transmission at a single synapse in the giant fiber (GF) escape circuit, which can be mitigated by reduced insulin signaling [15].

The core of the GF circuit is a pair of "giant" interneurons, which descend from the brain to the thorax where they form mixed electrical/cholinergic output synapses with the motor neuron of the tergotrochanteral "jump" muscle (TTM), and with an interneuron that contacts the dorsal longitudinal flight muscle (DLM) motor neuron [16–21] (Fig 1). It is the former of these output synapses that declines in efficacy with age [15]. An action potential in the GF is evoked by a combination of fast visual looming [22–24] and air movement [25,26], and subsequently elicits an escape jump. Rapid air movements (and sound) are detected by a subpopulation of antennal mechanosensory neurons in the Johnston's Organ (JO), the *Drosophila* analog of the mammalian inner ear [27] and, within the brain, these JONs form mixed electrical/chemical synapses onto the anterior GF dendrite [28–31].

These same *Drosophila* auditory neurons have at least one additional role, which is detection of the mating courtship song [33–35]. The song, produced by the male vibrating an extended wing, has two modes, sine and pulse, the latter being particularly important for mate selection [36–39]. These potentially conflicting functions of the auditory system (rapid escape versus song perception) may change in relative importance as the animal matures, mates and grows older, raising the possibility that, like the output of the GF, the input to the GF from the JONs perhaps also declines with age. There is a brief report that the amplitude of synaptic currents of auditory neurons recorded in the GF declines shortly after the adult ecloses [30], but this has not been pursued further.

In our previous studies we developed a way to measure simultaneously the amplitude of the compound JON action potentials and, albeit indirectly, the strength of the connection between the auditory inputs and the GF [28,29]. Here we describe a more direct assay that involves monitoring GF excitability via the neck connective, which allows long-term recording of GF responses to sound. We then use this technique to quantify the strength of the antennal response to sound, and of the auditory neuron—GF synapse, in males and female of different ages. We find that in both sexes the response to sound increases somewhat at around the time when mating is taking place then decreases with age, however, the strength of the connection between the auditory input and the GF itself remains unchanged.

## Materials and methods

### Flies

*Drosophila melanogaster* flies of the *Canton-S* genotype were obtained from Dr. Andrew Seeds (originally from Dr. Martin Heisenberg). For poisoning the GF, the *GAL4* driver line *R68A06-GAL4* (39449) [22] was obtained from the Bloomington Stock Center. The effector line used was *tub-GAL80^{ts}; UAS-DTI* {Katja Brückner [40]}. *GAL4* lines were crossed with the respective *UAS* lines and the F1 used for experiments. Flies were reared on cornmeal media and raised at 25°C and 60% relative humidity with males and females together until the required experimental age, changed to fresh food vials weekly.

### Electrophysiology

Flies were briefly chilled at 4°C before mounting on a microscope slide with dental wax by immobilizing the legs, then pushing up a wall of wax on each side of the fly and immobilizing

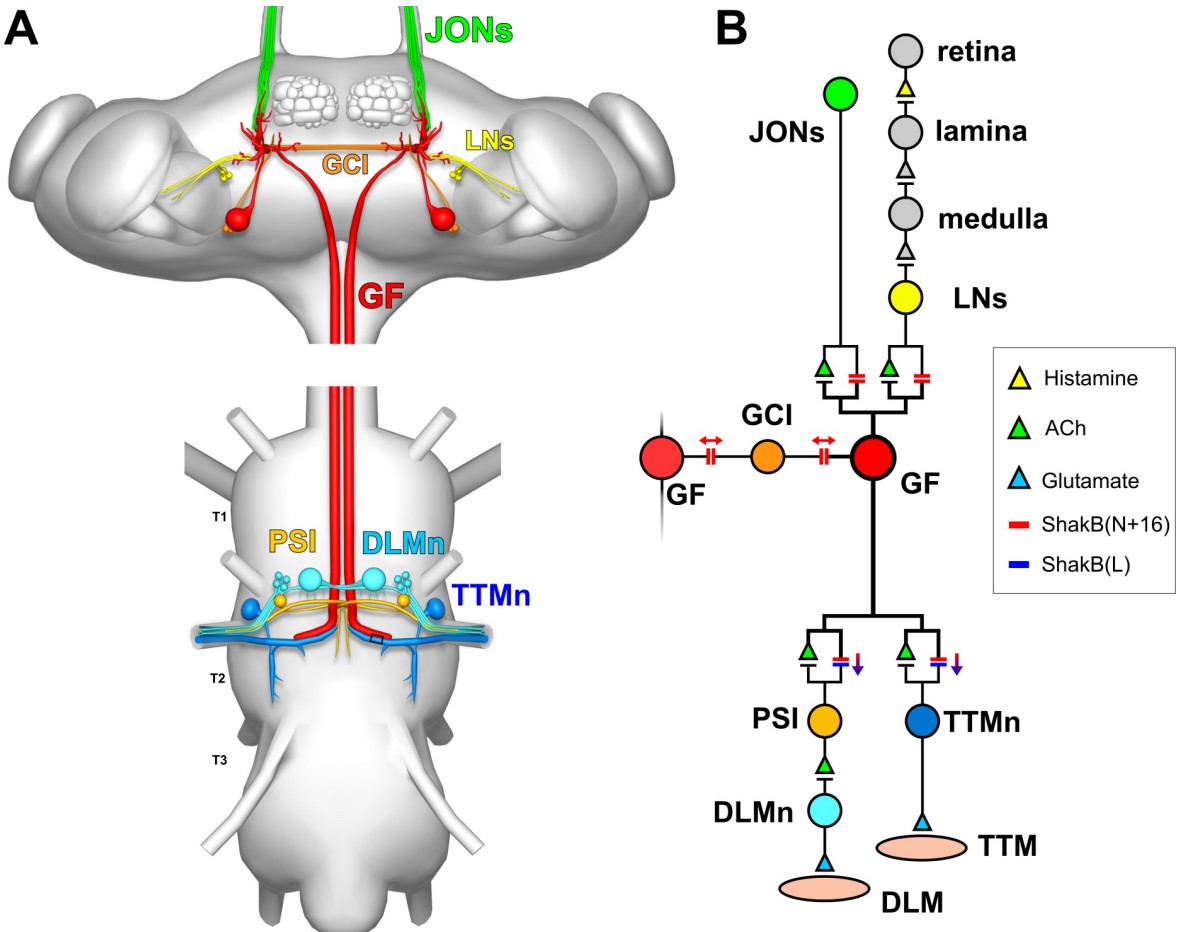

**Fig 1. The Drosophila giant fiber escape circuit.** (A) Anatomy of the system, viewed from dorsal. In the brain, the GF receives synaptic inputs onto its dendritic branches from the mechanosensory JONs and from polysynaptic visual pathways (LNs: lobular neurons). It also forms gap junctions with the giant commissural interneurons (GCI). The GF axon descends to the thoracic ganglion where it forms electrical and chemical synapses with the tergotrochanteral motorneuron (TTMn) of the TTM jump muscle, and the peripherally-synapsing interneuron (PSI) which innervates the dorsal longitudinal motorneurons (DLMn) of the DLM indirect flight muscles [32]. (B) Wiring diagram of the main components of the system. Chemical synapses are denoted by triangles, colored to represent the transmitter where it is known. Electrical synapses are denoted by double bars, colored to indicate which Shaking B (ShakB) innexin isoforms they are composed of, with arrows indicating putative rectifying or non-rectifying junctions [18,32].

the wings. A short piece of human finger hair was used on each side to push forward the head slightly so that the neck was visible (Fig 2). The fly was transferred to a platform under a dissecting microscope (see [41]) and a grounding electrode was inserted into the abdomen. All electrodes were made from electrolytically sharpened tungsten wire [32,37]. Six additional tungsten electrodes, mounted on miniature Narishige micromanipulators, were subsequently inserted in the sequence, and positions, illustrated in Fig 2. The first stimulation electrode, connected to the positive output of the SIU5 Isolation Unit of a S48 stimulator (Grass Technologies, RI, USA) was inserted shallowly in the right eye, then the second was inserted in the left eye. Together, the stimulation electrodes were moved anteriorly so that the neck was held in a slightly stretched position, exposing the anterior of the antepronotum (Fig 2). The third, antennal indifferent electrode was inserted shallowly into the right-hand side of the head capsule, midway between the middle orbital bristle and the ocellar bristle. The fourth, connective indifferent electrode was inserted shallowly into the posterior left side of the neck, close to the

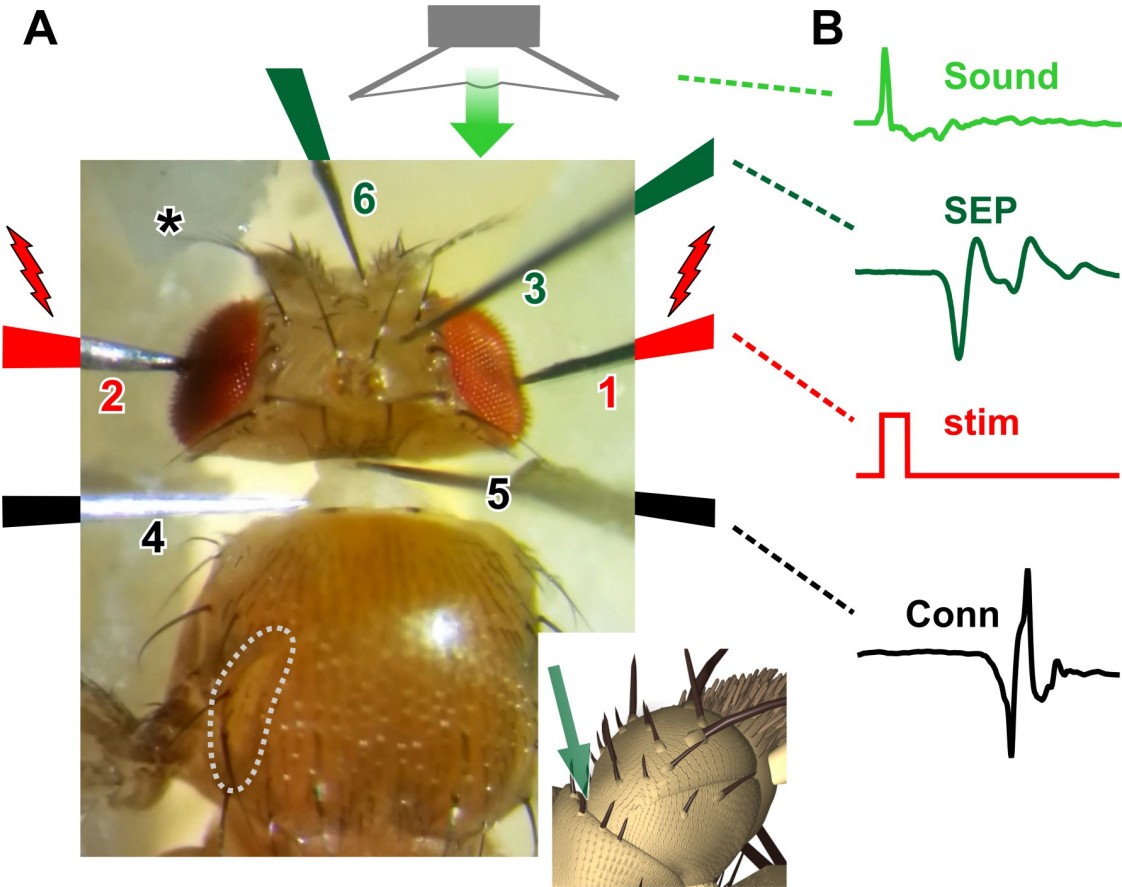

**Fig 2. New recording system.** (A) Photograph of the recording setup showing, in order of insertion, the electrolytically-sharpened tungsten electrodes. 1. First stimulation electrode in the right eye. 2. Second stimulation electrode in the left eye. 3. Antennal indifferent electrode inserted midway between middle orbital bristle and the ocellar bristle. 4. Indifferent electrode inserted in the posterior left side of the neck, close to the antepronotum. 5. Neck electrode inserted anteriorly, just behind the occipital bristles at the midline. 6. Antennal electrode inserted into the right inter-pedicel-scape cuticle (as indicated in insert below). The left antenna is immobilized with a drop of high-vacuum grease (asterisk). The antennal auditory neurons are stimulated by a brief sound pulse from a loudspeaker, delivered via a tube. (B) The neuronal responses are recorded from the antennal nerve in the form of a compound action potential, or sound-evoked potential (SEP). The visual pathway to the GF is stimulated with a 1 ms current pulse (stim) which, when suprathreshold, results in a characteristically-shaped biphasic action potential recorded from the neck connective (Conn), and also a visible twitch in the thoracic TTM muscles, the insertion of one of which is outlined (gray dotted line in A).

antepronotum. The fifth, recording connective electrode was inserted into the anterior midline of the neck just behind the occipital bristles, and deeply enough so as to penetrate the esophagus but not damage the underlying neck connective. The sixth and final (antennal) electrode was inserted into the inter-pedicel-scape cuticle on the medial side of the right-hand antenna (see insert in Fig 2). The left antennal arista was immobilized with a drop of high-vacuum grease. Preparations were discarded if the right antennal pedicel and arista were not able to move freely in response to a light buccal puff of air. Experiments were carried at the laboratory temperature of 19° C. Videos and stills were taken with a Samsung 5 cellular telephone mounted over one microscope ocular.

The signals were amplified x10000 using a differential AC 1700 amplifier (A-M Systems, WA USA) and a Brownlee Precision 210A (Brownlee Precision Co., CA USA) amplifier, bandpass filtered between 10 Hz and 20 kHz, notch-filtered at 60 Hz, digitized with an Axon Instruments Digidata 1550B (Molecular Devices, LLC, CA USA), and acquired and sampled at 50

kHz with pClamp 10 (Molecular Devices). Traces were analyzed with Clampfit 10 (Molecular Devices), or WinWCP (Strathclyde Electrophysiology Software) from which images were imported into Coreldraw 10 (Corel Corp., Canada).

The antennal auditory neurons were stimulated by a brief sound impulse, or 'click', generated with an abrupt-onset half-cycle sawtooth wave (rise time 0.05 ms, decay time 25 ms) in the pClamp software, passed via the Brownlee Precision amplifier then a MPA-50 40 Watt PA amplifier (Radio Shack) to an Optimus loudspeaker placed approximately 50 cm distant, facing away from the preparation so as to avoid artifacts generated by loudspeaker coil movements. As described in other studies, near-field sound was delivered from the loudspeaker to within about 1 cm of the antenna via a 1 cm diameter Tygon tube approximately 50 cm in length [42]. A small tweeter speaker placed just in front of the main loudspeaker near the entrance of the tube was connected to the Brownlee Precision amplifier and served as a simple dynamic microphone to monitor the time course and relative amplitudes of the sound impulse stimuli. The large first peak of the sound signal was approximately 0.5–1 ms in duration (from onset to zero-crossing) and its amplitude was measured in arbitrary units (Fig 2). Sound pressure levels (SPL) were estimated at the tube entrance with a RadioShack digital sound meter, and ranged from about 73 dB to 111 dB (125 ms time constant, Z weighting). The *Drosophila* antenna is a near-field sound receptor, responding best close to the sound source (within one wavelength), where there is bulk air particle displacement [34]. This close to the source the sound particle velocity is not directly proportional to the sound pressure gradient. However, equipment to calibrate the actual sound particle velocities was not available, so the sound amplitudes detected by our loudspeaker/microphone are presented here in arbitrary units. Since we are only interested in relative comparisons between different ages this has no effect on the results.

Antennal nerve responses were recorded from the nerve in the form of a compound action potential, here termed a "sound-evoked potential" (SEP) [31,34,42] (Fig 2). GF activity was monitored directly. Previously, we used the traditional method of monitoring GF activity indirectly, using muscle action potentials in TTM and DLM as a reliable readout for GF spikes [28,41]. Here we recorded compound spiking activity directly from the neck connective, allowing a characteristic biphasic GF spike signal to be detected (see Results for its characterization). Monitoring GF responses to sound required our previous indirect summation technique [28]. Even with the most intense sound impulse stimuli, which should be optimal for evoking activity in the A group of JONs that connect directly to the GF [43], an action potential in the GF could not be obtained. It has long been known that dipteran giant descending neurons (of which the GF is an example) respond both to air puffs/antennal movement as well as stimulation of optical pathways [44], so we again adopted the strategy of electrically stimulating the optical pathways but keeping them subthreshold, and allowing them to summate with the sub-threshold auditory inputs, thus eliciting GF activity [28]. To this end, the visual pathway to the GF was stimulated with a 1 ms current pulse across the eyes which, when suprathreshold (> 3V), resulted, about 4 ms later, in a characteristically-shaped biphasic action potential recorded from the neck connective electrodes, and also a visible twitch in the TTM muscle, the insertion of one of which is outlined in Fig 2. This corresponds to the previous-described long latency response that can be recorded in the TTM and DLM muscles, and which results from activation of the polysynaptic (3–4 synapses) visual pathway [45–47].

## Statistics

Data from these experiments are presented in S1 File. N represents the number of animals. The normality of the distribution of the data sets was first determined, and subsequent tests

were carried out, using PAST3 software [48]. To identify significant differences between means of control vs. experimental groups, normally-distributed data were compared with ANOVA followed by *post-hoc* Tukey tests, whereas non-normally distributed data were compared with a Kruskal-Wallis test followed by Mann-Whitney pairwise comparisons with Bonferroni-corrected p values. In figures, * denotes $p \leq 0.05$, ** $p \leq 0.01$, *** $p \leq 0.001$. Plots were made with Excel and transferred to CorelDraw for construction of the graphs.

## Results

### JONs become less sensitive to sound with age

JONs were stimulated with a transient sound "click", as in an earlier study [30]–this brief intense sound impulse (first peak approximately 0.5 to 1 ms duration) should represent an ideal stimulus for that subset of JONs that directly contact the GF dendrite [28], and which respond maximally to transients or high frequency tones [43]. Since the main point of this study was to maximally stimulate the GF in as short a time as possible, a more extensive repertoire of sound stimulus patterns, frequencies etc. was not used. SEPs in the antennal nerve were first detectable at a sound level of 1 (arbitrary units) with a peak amplitude of 0.05–0.1 mV and a latency of approximately 2.8 ms (Fig 3A). This sound level, measured right next to the loudspeaker at about 73 dB SPL, probably corresponds to an air velocity at the antenna of about 6 x $10^{-5}$ m/s, assuming our Canton-S animals have comparable auditory acuity to the "Dickinson wild-caught" animals used in an earlier study [30]. The amplitude of the (negative) SEP peak increased with sound level to reach peak amplitude by level 30–50 (approximately 105 dB SPL, equivalent to about 6 x $10^{-3}$ m/s) (Fig 3B), with a latency of 1.8 ms. It can be calculated that, with the speed of sound being 34 cm/ms, 1.47 ms of this latency is due to the 50 cm length of the air delivery tube.

Variability of the maximum SEP amplitude was quite high between individual animals; for example, Fig 3B illustrates an extreme case in the responses of seven 10-day-old females, which exhibit maximal SEP amplitudes ranging from 0.53 to 1.72 mV. This amount of variation has been reported by other laboratories [42], and in our case could be partially due to variations in experimental conditions, techniques, or electrode placement, although an attempt was made to maintain these constant throughout. The local field potential that makes up the SEP is a compound action potential that presumably represents the sum of all the currents of the JONs that are spiking synchronously. Biological variation in the size of this signal is likely due to individual differences in the total number of JONs that spike in response to sound, and/or to variation in the timing or kinetics of the JON action potential currents. Because of this variation, the SEP amplitude was normalized to the maximum response, as illustrated for the seven animals in Fig 3C. For each animal the sound level at which the 50% maximal SEP was evoked was obtained from this plot, as illustrated. In the particular case of these 7 females (which was fairly representative of the population in general), these half-maximal sound levels ranged from 2.4 to 5.1, with a mean of 3.85 ± 0.31.

Sound levels that elicited half-maximal SEPs were calculated from experiments such as those shown in Fig 3C for male and female Canton-S animals at different ages (in days): 1, 10, 20, 30, and 50 (Fig 4). At no age was there any significant difference between males and females, so these data were combined. There was a significant 40% decrease in the mean sound level that elicited a half-maximal SEP (i.e. an increase in sensitivity) from 6.90 ± 0.63 to 4.16 ± 0.42 between 1 and 10 d (Fig 4). Between 10 and 20 d the mean sound level that elicited a half-maximal SEP increased significantly back up to 7.60 ± 0.74, remained constant in 30 d animals, then increased further at 50 d (compared to 1d or 10 d) to 11.57 ± 1.57, although there was wide variation in the 7 animals tested. The results indicate that males and females

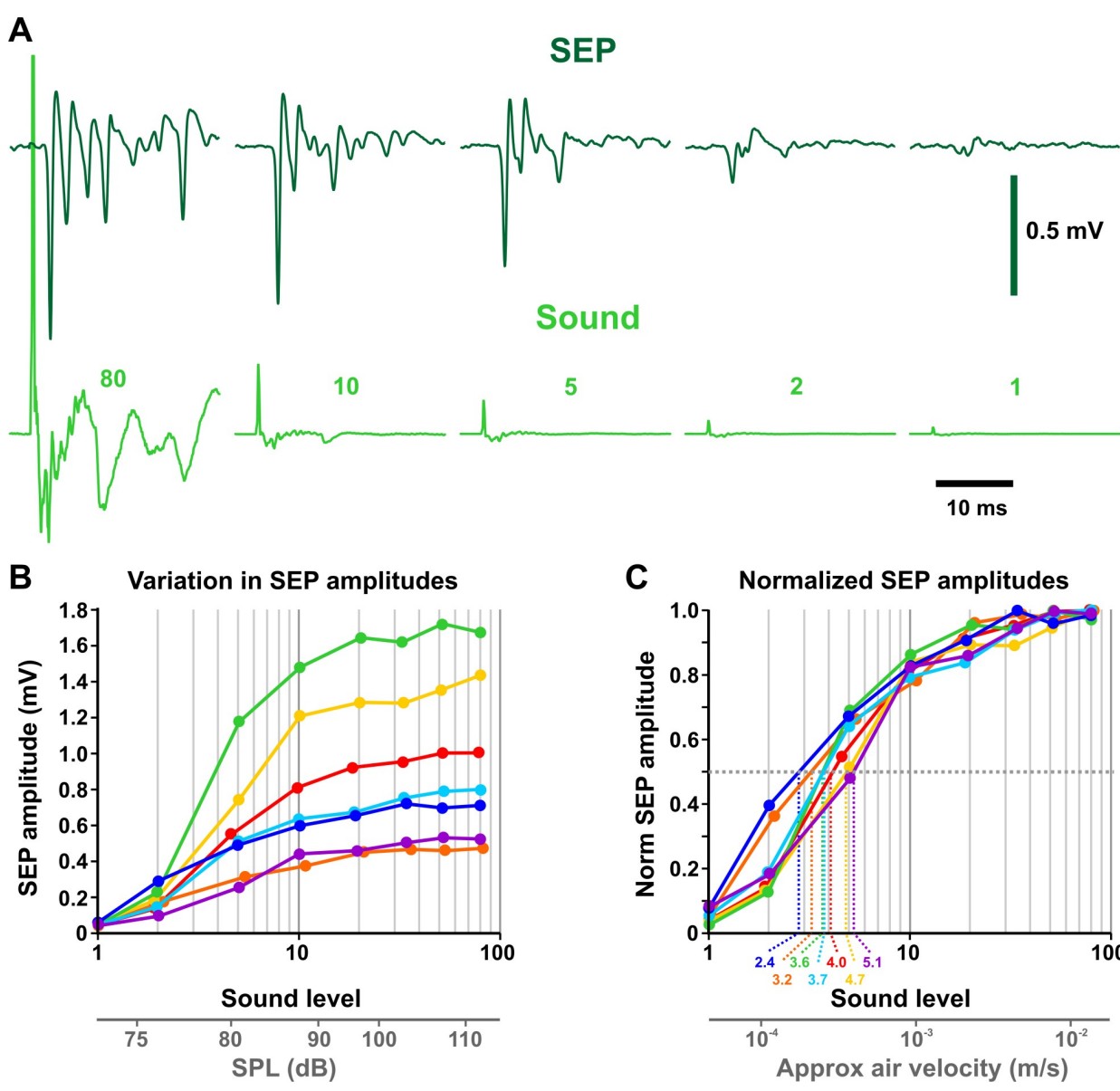

**Fig 3. Responses of antennal neurons to sound.** (A) Sound-evoked potentials (SEPs) recorded from the antennal nerve in response to sound impulses of different magnitudes. Average of 20 traces. The sound signal was measured next to the main loudspeaker with a small tweeter attached to an amplifier. The amplitude of the first peak of the sound impulse is shown over each trace in arbitrary units. (B) Responses (SEPs) of antennal auditory neurons to sound impulses of different magnitudes (sound level), from 7 different females at 10 days of age, showing a marked variation in peak SEP amplitude, from 0.5 to 1.7 mV. The response latency increases as sound level decreases, with a latency of 1.8 ms for intensity 80, and a latency of 2.8 ms for intensity 1. Sound level is expressed as arbitrary units on a log scale. Below is shown the approximate equivalent sound pressure level (SPL) in dB. (C) Normalization of the recordings to the maximum amplitude. Dashed lines indicate the sound levels that elicit a half-maximal SEP in each preparation. Below is shown the approximate equivalent air velocity in m/s, obtained from [30].

are thus maximally sensitive to transient sound impulses between 1 and 10 d of age and become less sensitive (i.e. "deafer") as they age, with very old 50 d flies being almost 3x less sensitive than at 10 d.

It has been reported that the amplitude of synaptic currents of auditory neurons recorded in the GF declines shortly after the adult hatches [30]; we did not test this in our system because the soft cuticle of newly-eclosed adults makes insertion of recording electrodes

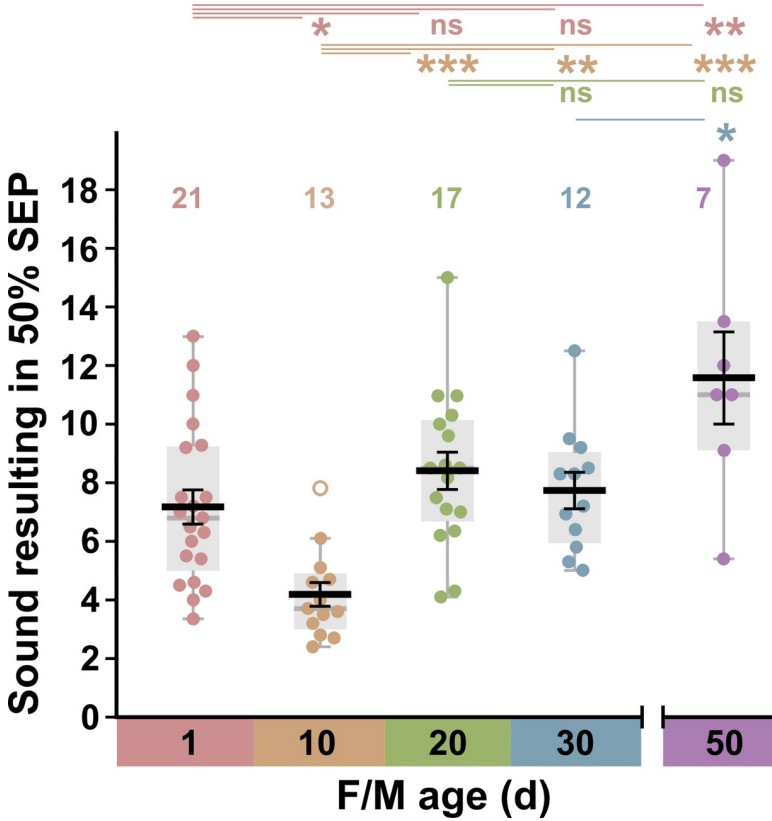

**Fig 4. Changes in antennal sound responses with age.** Sound levels that elicit a half-maximal antennal response, shown as box (light gray) and scatter plots of data, with superimposed means +/- SEM (black) and medians (gray bar), for grouped males and females of different ages. Outliers are shown as empty circles. Above are shown the results of ANOVA followed by *post-hoc* Tukey tests. There is a significant increase in antennal sensitivity to sound between 1 d and 10 d, followed by a halving of sensitivity between 10 d and 20 d. Antennae of aged animals at 50 d are approximately 3x less sensitive to sound than at 10 d.

difficult, and often leads to nerve damage. Female *Drosophila melanogaster* are already sexually receptive by 1 d post-eclosion [49] and attain peak fertility at 1 week, however the receptivity of virgins declines to 50% by 20 d and is almost zero at 40 d [50,51]. Males are not fully sexually mature until 3–4 d [52] and suffer a steep decrease in mating success between 35 and 42 d [53]. Thus our result that between 1 and 10 d the antennae are at their most sensitive to sound impulses is consistent with this being within the age range within which most matings take place, and therefore the time when detection of the mating song, and/or the ability to escape from predators, would be most crucial for the animals' fitness.

## GF action potentials recorded from the neck connective

In our previous studies we used an indirect method to monitor the spiking activity of the GF. First developed at least 40 years ago [45,46], that method takes advantage of the highly reliable electrical/chemical synaptic connection between the GF and its postsynaptic motor neurons: the TTMn and, via the peripherally-synapsing interneuron or PSI, the DLMn [41] (see Fig 1). Thus, an action potential in both of their downstream muscles (that of the DLM lagging 0.5–1 ms behind the TTM) indicates that the GF must have spiked. This method has been routinely used in a plethora of physiological, developmental, and pharmacological studies from many labs, for example [15,32,41,54–57], but we recently adapted it to monitor GF excitability in

response to auditory inputs [28,29]. Intracellular recordings from the GF soma can directly detect antennal synaptic inputs in the form of unitary postsynaptic potentials or currents [24,33,58–60], but very long term recordings and, more importantly, simultaneous measurement of the presynaptic antennal action potentials, are technically challenging. We therefore focused on detecting GF spikes in the neck connective with extracellular recordings, a technique which has been used occasionally by other laboratories [26].

Using the electrode placement shown in Fig 2, we were able to detect a small downward signal at about 4.5 ms with subthreshold stimuli across the eyes (Fig 5A). Excitatory visual input to the lateral dendrites of the GF is provided mainly by 55 lobular columnar type 4 (LC4) and 108 lobular plate/lobular columnar type 2 (LPLC2) neurons which, in turn, receive inputs from the medulla [24,60,61]. This small signal in the neck connective may represent the activity of some or all of those presynaptic lobular neurons, although they are apparently non-spiking [61]. Alternatively it could also perhaps represent the synaptic currents from these inputs in the GF itself.

With slightly larger, suprathreshold, stimuli a stereotypical biphasic signal appeared (Fig 5B). In 1-day-old animals (males and females) this signal was 0.22 ± 0.10 mV peak-to-peak amplitude and 0.64 ± 0.06 ms peak-to-peak duration, with a stimulus threshold of 2.71 ± 0.17 V (N = 19). These parameters did not change with age, although threshold decreased slightly at 10 d (2.37 ± 0.07 V, N = 12), and was higher again at 30 d (2.86 ± 0.15 V, N = 12). The biphasic signal we observe is consistent with it being the approximate differential of the intracellular action potential, with the negative portion representing the current from the rising phase and the positive part the falling phase [62]. The duration of a single GF action potential elicited by transocular stimulation and recorded intracellularly from the connective near the thorax is reportedly 0.40 ± 0.06 ms [63], see also [64], and approximately 1 ms when evoked by optogenetic activation of LC4 lobular neurons and recorded from the soma [24]. In addition, long duration (50 ms) $Ca^{2+}$-mediated action potentials can be elicited by somatic stimulation of GF [22,33], but their physiological relevance is unclear and we did not observe them in our experiments.

An alternative possibility that can be discarded is that the delay between the negative and positive phases of the extracellularly-recorded GF action potential reflects the time that the signal takes to travel between the two electrodes. This is unlikely because the maximum transmission delay between direct stimulation of the GF and excitation of the TTMn is 0.52 ms in immature GFs [57], and the distance between our electrodes is less than one quarter of that between brain and the GF-TTMn synapse. The two GFs are electrically coupled together [19] and thus thought to fire synchronously–this may account for the somewhat longer duration of the extracellular signal (0.64 versus 0.40 ms) that we observe. As a final confirmation of its identity, targeted genetic poisoning of the GFs by driving diphtheria toxin [65] with *R68A06-GAL4* [66,67] removed the characteristic biphasic action potential signal from connective recordings (Fig 5D). In these animals (N = 3) it appeared that the small subthreshold signal remained, indicating that it did not originate in the GF itself (see above). While we did not verify that the GFs were completely absent in these animals, their normally prominent axons [28] were not visible when the neck connective was inspected with Nomarski optics (N = 2).

## Responses of the GF to sound

As before [28,29] we used a subthreshold stimulus of the visual circuits to summate with antennal nerve auditory input and bring the GF to threshold, since it will not spike in response to the latter alone. It has been shown that strong optogenetic stimulation of the presynaptic LC4 neurons is sufficient to evoke a GF action potential [24] so it is likely that our current pulse indirectly stimulates these, and other, lobular neurons. Using an eye stimulus delay of 1

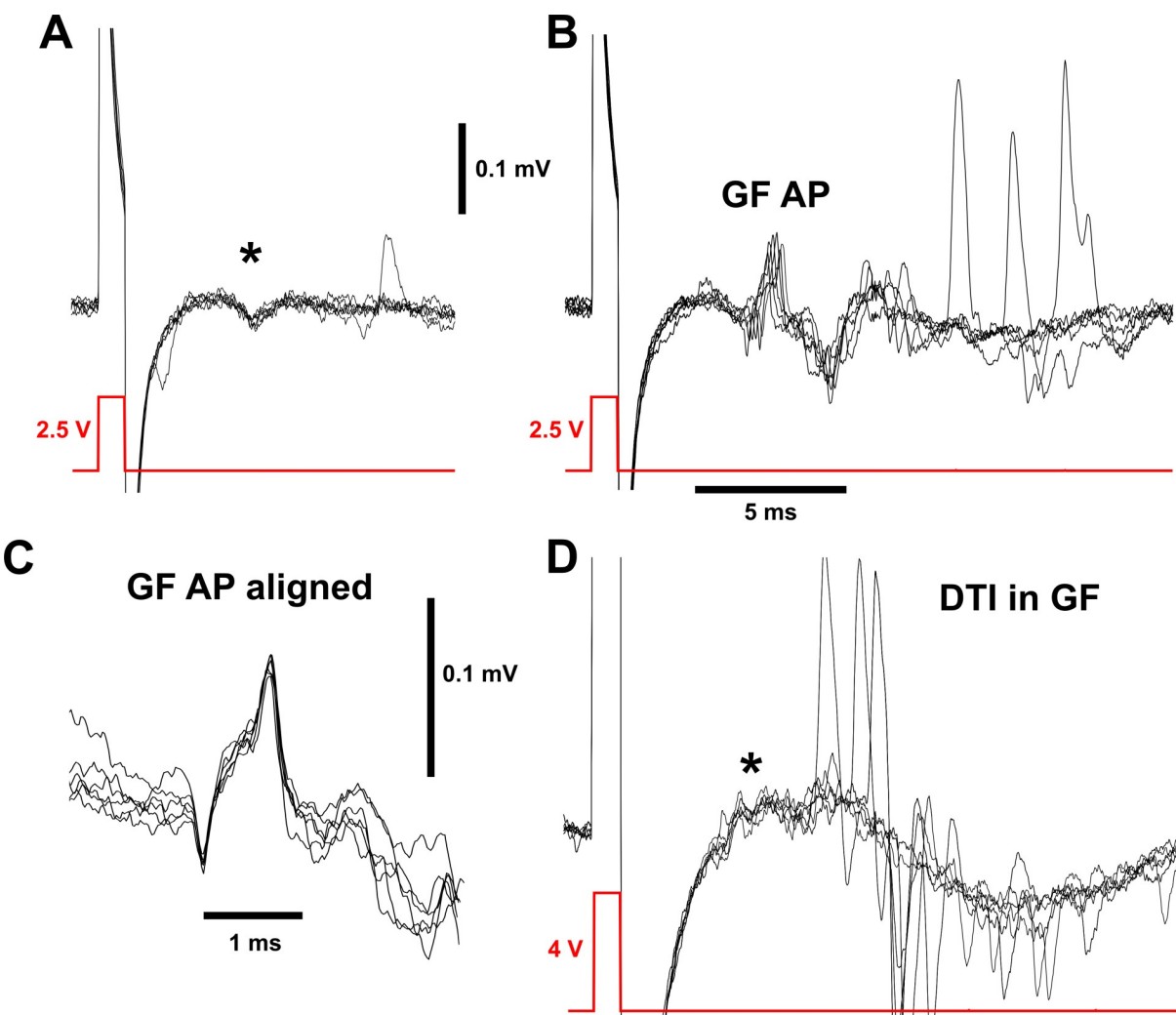

**Fig 5. Biphasic GF action potential recorded extracellularly in the connective.** (A) Six overlaid recordings from the neck connective, showing small responses (asterisk) to a 2.5 V stimulus pulse (just subthreshold) across the eyes. (B) In 6 of 20 instances, the same stimulus passed threshold and gave a characteristic biphasic signal with 5 ms latency that represents the GF action potential (GF AP). Other, larger signals are sometimes observed in the connective, at variable latencies. These presumably originate from other, unidentified neurons or perhaps muscles. (C) The GF signal, enlarged with all traces aligned, to show the stereotypical biphasic shape. (D) Recording from the connective of an animal of genotype *tub-GAL80^ts/+; UAS-DTI/R68A06-GAL4*, showing the absence of the characteristic GF AP with what would normally be a suprathreshold stimulus of 4 V, but the apparent persistence of the small subthreshold signal (asterisk). Large, unidentified, signals that are not time-locked to the stimulus are still present. Stimuli of 6 V were also used but were not shown due to the large stimulus artifact.

ms from trace onset (Fig 6A), so as to precede by 9 ms the sound impulse, the voltage was adjusted until the GF spiked in approximately 50% of the trials ("threshold"), then decreased in 0.05 V increments until GF spike probability was 1/20 or less. The eye stimulus delay from trace onset was then increased to approximately 9 ms so that the small subthreshold response in the connective (see above) was synchronous with the antennal nerve SEP, and the probability of GF spiking was again measured over 20 trials (Fig 6B). During the course of an experiment, eye stimulus delays of 7.5 to 10 ms from trace onset were assayed in 0.5 ms steps to find the most effective for stimulating the GF. Generally for large amplitude SEPs a stimulus delay of 8.0–8.5 ms was optimal, whereas for low-intensity sounds and the resulting slightly delayed SEPs (see above) a 9.5–10.0 delay was required.

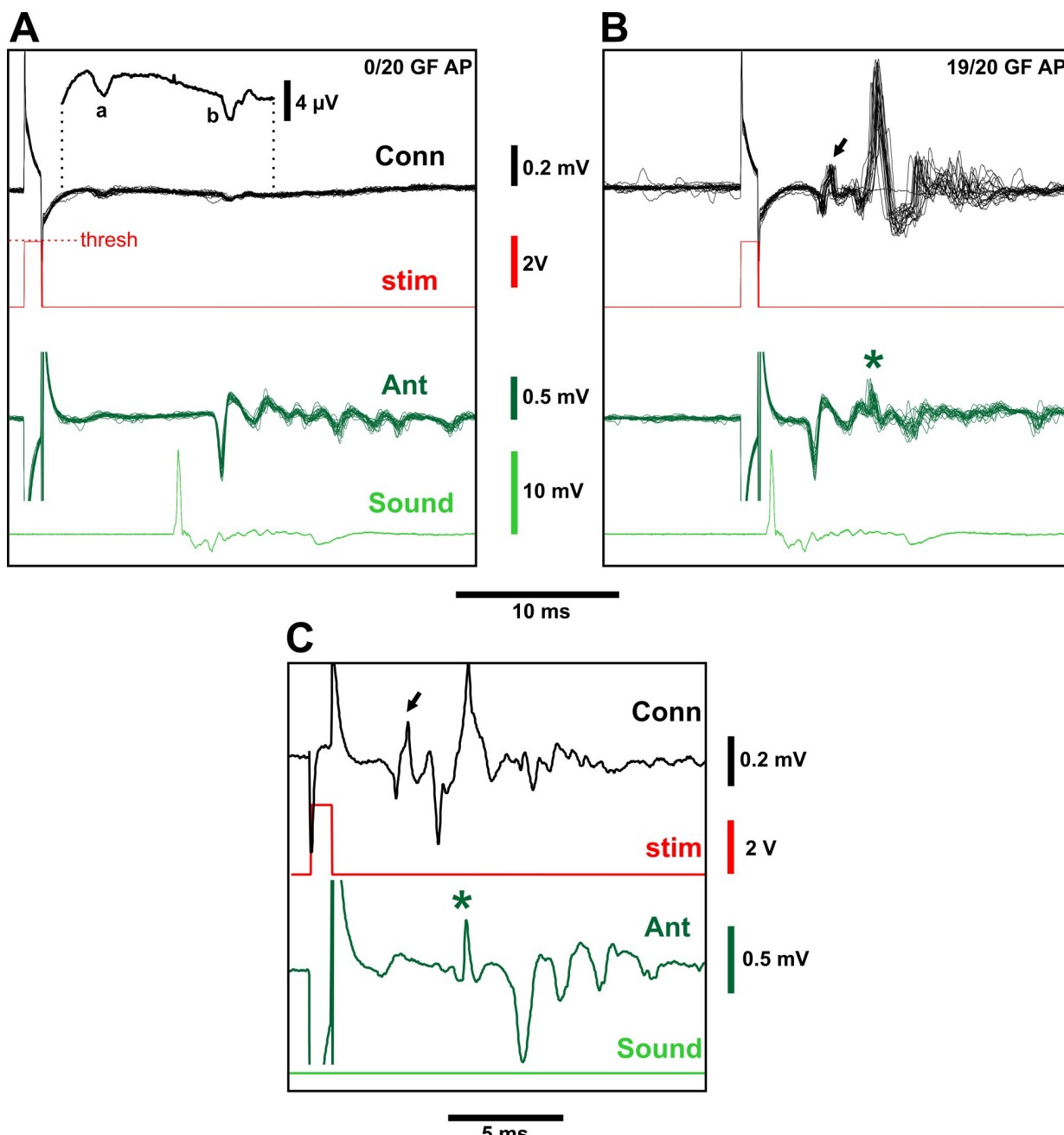

**Fig 6. Responses of GF to sound.** (A) Twenty overlaid recordings from the connective and antennal nerve, showing an SEP in the latter in response to a sound impulse. The preceding subthreshold eye stimulus gives no GF spike, but both it and the sound result in small downward responses ("a" and "b" in the magnified average of the 20 traces shown above). (B) The same preparation with the subthreshold eye stimulus synchronized to the sound impulse. Less than 1 ms after the SEP, there is a characteristic biphasic GF spike in 19 of the 20 traces (arrow), followed, a few ms later, by large potentials from unidentified neuronal and muscular activity. Some of this later activity is also present in the antennal nerve trace (asterisk). (C) A single trace from a different experiment with no sound impulse, illustrating how the GF action potential (arrow) is immediately followed by a depolarization in the antennal nerve (asterisk) and a twitch of the antenna, which then results in a SEP-like response even when there is no sound. The stimulus artifacts in the antennal traces have been truncated for clarity.

In addition to the small response in the connective to subthreshold eye stimuli ("a" in Fig 6A), we also observed a small connective response to sound ("b" in Fig 6A). The latter lagged behind the antennal SEP by approximately 0.5 ms, and increased linearly with the percentage

SEP amplitude. This small connective SEP reached a maximum of about 0.01–0.02 mV. One intriguing early possibility was that it represented a subthreshold response to sound in the GF itself, however, it was still visible in *tub-GAL80^{ts}/+; UAS-DTI/R68A06-GAL4* animals, suggesting that this is not the case. The small connective SEP ("b" in Fig 6A) most likely represents a greatly attenuated signal from the antennal nerve spiking activity, since one connective electrode is positioned more anteriorly than the other allowing for a differential antennal signal. The fact that it also echoes the multi-peaked antennal response to a tone stimulus (S1 Fig) supports this idea.

A complex pattern of activity in the connective was invariably observed 2–3 ms after the GF action potential, consisting of one or two large, and several smaller, action potentials (Fig 6B). One of these also coincided with an additional non-SEP spike in the antennal recording, which correlated with a visible backwards twitch of the antenna (S1 Video). In the complete absence of sound, a suprathreshold stimulus of GF also elicited this antennal twitch, which in turn was sufficient to stimulate the JONs and produce an 'illusory SEP' (Fig 6C). Two opposing antennal muscles move each scape-pedicel joint in response to visual motion [68], but it has not been previously reported that these are activated directly or indirectly by the GF.

## The JO input to the GF does not change with age

Using the method described above, an input-output curve was constructed for each GF, measuring the probability of its spiking in response to normalized SEPs of different amplitudes (Fig 7A). Both sexes were used, with ages of 1, 10, 20, 30 and 50 d, however, males and females were not significantly different and so their data were grouped together. From each curve we obtained the normalized SEP amplitude that resulted in a 50% GF spike probability. These data are shown in Fig 7B, which shows that the JON-GF synaptic connection was quite variable

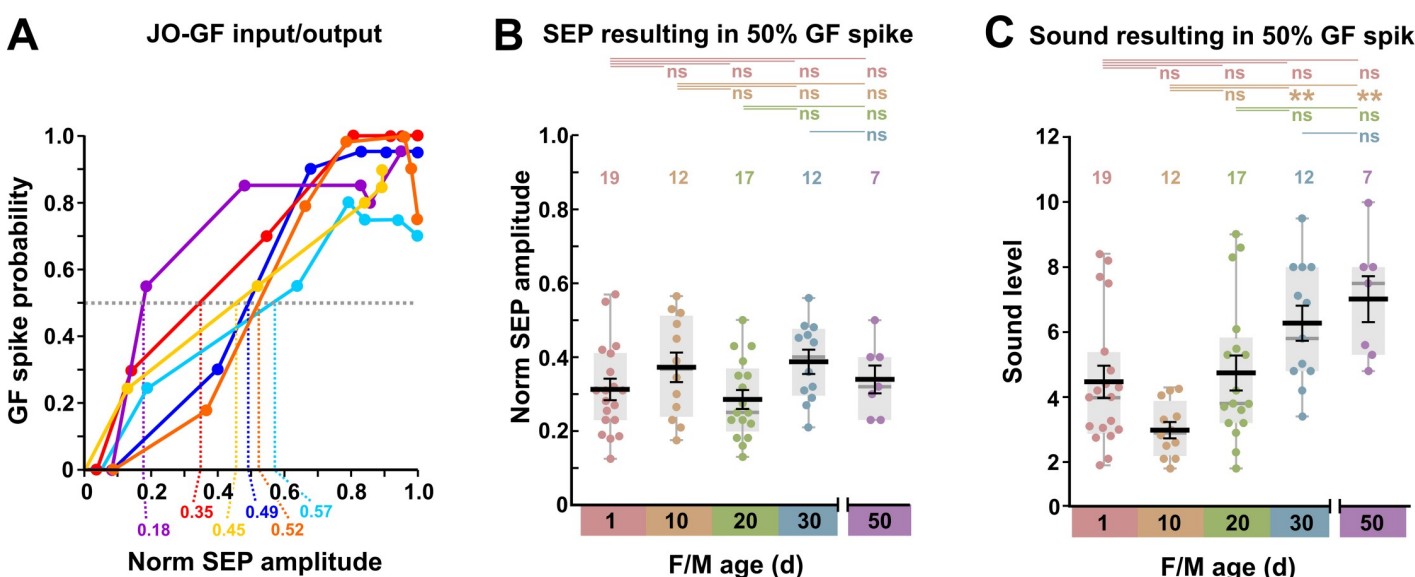

**Fig 7. Changes in GF sound responses with age.** (A) The probability of GF producing an action potential (spike) for SEPs of different normalized amplitudes, in 6 of the preparations shown in Fig 3. Dashed lines indicate the normalized SEP amplitudes at which the GF spiking probability was 50%, which become data points in B and C. (B) and (C) Box and scatter plots of data, with superimposed mean +/- SEM, for grouped males and females of different ages (d). NB. There was no significant difference between males and females at any age. (B) There is no significant change in the strength of the synapse between the auditory neuron and the GF, i.e., the SEP that results in a half-maximal GF spike probability does not change. Above are shown the results of ANOVA followed by *post-hoc* Tukey tests. (C) Because the antennal sound sensitivity changes but the synapse strength remains constant, there is a corresponding decrease in sound sensitivity of the GF between 10 d and 30–50 d. Above are shown the results of Kruskal-Wallis followed by Bonferroni-corrected Mann-Whitney pairwise tests.

in strength at all ages. Different animals showed a 50% GF spiking probability at normalized SEP amplitudes ranging from 0.13 to 0.58. There were thus no significant changes in the strength of the JON-GF synaptic connection with age.

Finally, to quantify the response of the GF to sound, we plotted the sound level that resulted in a 50% GF spiking probability (Fig 7C). These distributions are somewhat similar to those in Fig 4, however, because of the variability in JON-GF synapse strength the only significant differences are those between day 10 and days 30 and 50.

## Discussion

Some logical inferences can be made from these data about the relative amplitudes of lobular neuron (LN) and JON postsynaptic potentials (PSPs) in GF. The LN PSP can be driven to surpass threshold, unlike the JON PSP. (1) By adjusting the stimulus pulse, the LN PSP can also be reduced to (and below) the level where only 1/20 action potentials are produced, at which point, if its amplitude distribution is approximately normal, the mean will be 2 standard deviations (SD) below threshold. Thus, for an action potential threshold of 10 mV, which is a reasonable estimate as measured from soma recordings [24,60], the LN PSP could, for example, have an amplitude of 4 ± 3 mV (mean ± SD), alternatively 6 ± 2 mV, or (less likely) 8 ± 1 mV, etc. (2) Conversely, in the situation where LN and JON PSPs summate and evoke at least 19/20 action potentials (as was often the case), the summated PSP will have a mean amplitude of 2 SD above threshold and, (3) assuming linear PSP summation, this amplitude should equal the sum of LN and JON PSPs. (4) The variance of the summated PSP is the sum of the variances of LN and JON PSPs. (5) Finally, since JON PSPs alone cannot elicit GF action potentials, maximum JON amplitude is constrained to be at least 3 SD below threshold. There are several solutions that fulfill these five criteria (S2 File) but the most likely, in which the SDs are not unrealistically small, and where the JON PSP is smaller than the LN PSP, range from LN PSP = 7.0 ± 1.50 mV (mean ± SD) and JON PSP = 6.61 ± 1.09 mV; to LN PSP = 8.5 ± 0.75 mV and JON PSP = 5.09 ± 1.63 mV.

Unitary PSPs of approximately 0.5–1 mV can be recorded from the GF cell body and represent mechanosensory input from spontaneous JON action potentials [24,30,59,60]. An upper limit of 145 for the number of JONs providing auditory input to the GF has been provided [30], although the actual recordings shown in that paper indicate that this could be an overestimate by a factor of 10. We have previously determined the number of JO-A JON axons that cluster around the GF anterior dendrite, and that are electrically coupled to it, is 12–13 [28], although some few JO-B axons may also form additional chemical contacts (of uncertain functionality) with that dendrite's side branches [69]. Our estimates of maximal JON PSP amplitude are thus consistent with these estimates (13 x 0.5 mV unitary PSPs).

There is a concern that our conclusions regarding the input-output relationship of the synapse will also be affected by the unavoidably indirect nature of the presynaptic activity measurement. The compound field potential that makes up the SEP is the result of summation of all the action potentials from JONs that are firing in synchrony in response to the sound. This synchrony is promoted by the gap-junctional coupling between JONs [70]. A problem would arise if the sound-responsive JONs only made a minor contribution to the total SEP amplitude. However, the stimulus that was used here (a brief 'click') is designed to stimulate only certain JON subgroups, namely the sound-sensitive A and B groups, and predominantly the former [43,71,72]. Different subgroups of JONs have been tested for their contribution to the SEP by ablating them with toxins, or silencing them using expression of the inwardly-rectifying potassium channel Kir2.1, using different *GAL4* drivers [31]. Ablation of the A or B groups of JONs resulted in 50–60% reduction in the amplitude of the SEP compared to controls. Only the A

group is normally coupled via gap junctions to the GF [28]. Taken together, we can be confident that the normalized amplitude of the SEP is a fairly accurate representation of the presynaptic input to the JON-GF synapse. However, in the future it would be desirable to complement this subtractive strategy with one in which all JONs are silenced, then the activity in only A (and/or B) groups is rescued [30].

Our main conclusion is that, whereas JON sound sensitivity increases after eclosion, then decreases between 10 and 50 days of age, the strength of the JON-GF synapse itself remains constant, albeit rather variable. This result contrasts with that of the recent study which showed that the output synapse of the GF is decreased in strength at 45–47 d compared to 7 d [15]. In that study, the physiological measure of the decrease was an increased latency of transmission between GF and TTM, while the anatomical measure was a marked decrease in immunostaining for the innexin ShakB, which is required for the gap junctions at that synapse [16]. These deficits were rescued by reducing insulin signaling in the GF [15]. Thus it appears that the input synapses to the GF remain the same strength with age while its output synapses gets weaker, although it must be pointed out that the nearby output synapse of the GF with the PSI showed no such decrease in ShakB staining [15]. We can conclude that synapses formed by the same neuron can be differently affected by aging. In addition, the JONs may be less susceptible to senescence than the GF itself.

Our result raises the question of the molecular mechanisms for long-term maintenance of synaptic structure and function. Rapid turnover of their molecular constituents means that synapses are evanescent structures unless maintained [73]. The pattern, and strength, of a neuron's synaptic connections are critically important aspects of its identity and, as such, are maintained in part by the persistent expression of "terminal selector" transcription factors [74–76] via self-maintaining mechanisms such as transcriptional autoregulation [77]. It often seems to be the case that these transcription factors are used early in development for patterning the body or nervous system, and then are re-used later in life for maintenance of neuronal connectivity [76,78,79]. Whatever the molecular mechanisms for JON—GF synapse maintenance, keeping them operational in "middle-aged" flies is presumably adaptive, and therefore worth the metabolic costs, since males and females are still able to reproduce (albeit less successfully) until about 40 d of age [50,51,53].

## Supporting information

**S1 File. Experimental data from measurements of SEP amplitudes, GF firing probabilities etc, including the results of ANOVA and Kruskal-Wallis tests.**
(XLSX)

**S2 File. Simple Excel estimation of JON PSP amplitude distributions that summate to threshold with LN PSPs.**
(XLSX)

**S1 Fig. Small connective SEPs (Conn) follow the larger antennal SEPs (SEP) with a short tone stimulus.** NB the sound stimulus input itself is shown, not the resultant measured sound level.
(TIF)

**S1 Video. Video of a typical preparation, the same as that shown Fig 2.** Beginning at 7 s, suprathreshold stimulus pulses are passed into the eye electrode, at the same time as brief sound stimuli (audible in the background), resulting in GF action potentials. With each action potential the antennae twitch backwards and the TTM (only the left one is visible, slightly out

of focus) also contracts.
(MP4)

## Acknowledgments

We thank Andrew Seeds and Katja Brückner for donating fly stocks, and Carihann Dominicci for the loan of the microscope phone mount. Stocks obtained from the Bloomington Drosophila Stock Center (NIH P40OD018537) were used in this study.

## Author Contributions

**Conceptualization:** Jonathan M. Blagburn.

**Data curation:** Jonathan M. Blagburn.

**Formal analysis:** Jonathan M. Blagburn.

**Funding acquisition:** Jonathan M. Blagburn.

**Investigation:** Jonathan M. Blagburn.

**Methodology:** Jonathan M. Blagburn.

**Project administration:** Jonathan M. Blagburn.

**Visualization:** Jonathan M. Blagburn.

**Writing – original draft:** Jonathan M. Blagburn.

**Writing – review & editing:** Jonathan M. Blagburn.

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
