## [Decision Letter · Decision Letter 0]

25 Nov 2019

PONE-D-19-27717

A new method of recording from the giant fiber of Drosophila melanogaster shows that the strength of its auditory inputs remains constant with age

PLOS ONE

Dear Dr Blagburn,

Thank you for submitting your manuscript to PLOS ONE. After careful consideration, we feel that it has merit but does not fully meet PLOS ONE’s publication criteria as it currently stands. Therefore, we invite you to submit a revised version of the manuscript that addresses the points raised during the review process.

Line 28: Please, revise for clarity the sentence describing the 40% decrease and  the following few sentences.

Line 44: "motor neuron" or "motoneuron". Also line 284.

Line 49: I think you can delete the "somewhat uncoordinated"

Line 63: The shakeb should be designated as innexins and with a reference.

 Line 70. Eliminate quotations around love song and please describe the courtship song in a sentence or two for non-Drosophila readers

Line 84: Consider using "the strength of the connection between auditory input and GF"

Line 96: "Changes" in food probably does not mean different types of food? Please clarify in the text.

Please, explain at line 163  the relationship between sound intensity and particle velocity. Are they linear? Some incite for readers who is not familiar with particle velocity might be beneficial.

Line 164: Sound duration and frequency should be given. See also line 197: What does i.e. mean here? Were different frequencies used?

Line 166 and following: Maybe this could be improved a bit by explaining the general idea at the beginning of the paragraph: that an electrical stimulus activates optical pathways that are kept subthreshold, but enable the auditory stimulus to become suprathreshold and elicit GF activity. This would also help for later (line 384).

Line 208. Remove about. The latency should be directly measured and not roughly estimated.

Line 209 - 210. The phrase in the parentheses feels like a sentence fragment. Please revise it.

Line 267. Remove "in passing". It's either been reported or not.

Line 289. Remove the whole sentence stating that a direct measurement of GF activity would be more advantageous. It's unnecessary and such direct measurements have their own limitations (equipment expenses, more difficult, invasive, etc).

Line 350. 1ms delay to what?

Line 404 / 405: Maybe this sentence would be better if split into two.

Split sentence in line 406 into smaller statements.

Line 427: It's not clear why this is a combined results / discussion section. The discussion part clearly starts in line 427, so why not call it discussion here? (just a suggestion)

Line 441. This half sentence needs to be moved to line 433.

Figure 4. Remove the figure title. Perhaps swap it for the y-axis and use no figure title.

Figure 5. There are a lot of non GFN spikes in these recordings that don't seem reliably time locked to stimulation. Perhaps mentioning that these are from other non-identified neurons should be mentioned in the figure legend.

We would appreciate receiving your revised manuscript by Jan 09 2020 11:59PM. To enhance the reproducibility of your results, we recommend that if applicable you deposit your laboratory protocols in protocols.io, where a protocol can be assigned its own identifier (DOI) such that it can be cited independently in the future. For instructions see: http://journals.plos.org/plosone/s/submission-guidelines#loc-laboratory-protocols

We look forward to receiving your revised manuscript.

Kind regards,

Gennady Cymbalyuk, Ph.D.

Academic Editor

PLOS ONE

Journal Requirements:

Reviewers' comments:

Reviewer's Responses to Questions

**Comments to the Author**

1. Is the manuscript technically sound, and do the data support the conclusions?

Reviewer #1: Yes

Reviewer #2: Yes

2. Has the statistical analysis been performed appropriately and rigorously? 

Reviewer #1: Yes

Reviewer #2: Yes

3. Have the authors made all data underlying the findings in their manuscript fully available?

Reviewer #1: Yes

Reviewer #2: Yes

4. Is the manuscript presented in an intelligible fashion and written in standard English?

Reviewer #1: No

Reviewer #2: Yes

5. Review Comments to the Author

Reviewer #1: I enjoyed reading the manuscript by Jonathan Blackburn examining the changes in signal propagation through the giant fiber system over time in Drosophila. The paper introduces an elegant means of dissociating several synaptic steps in the giant fiber pathway by using sharp extracellular electrodes at several recording sites to deduce pre-synaptic activity in auditory and visual pathways from action potentials generated in the GFN. The data generally support the conclusion that JON sensitivity changes over the course of the flies lifetime while the giant fiber neuron response to auditory input remains constant. A small caveat is that high variability in the JON to GFN synapse strength may drown out small statistical effects. Essentially, it is difficult to prove there is no change over time as the change may simply be below statistical resolution. The study uses reasonable sample sizes to try to mitigate this risk. Overall, the study of synaptic aging is of high interest and this novel approach could facilitate identifying differences in synaptic maturation in a what that is more high throughput than direct synaptic physiology.

I do believe the manuscript could be improved eliminating some of the more informal writing style and reducing the use of quotations. Below are some additional specific suggestions.

1) Line 70. Eliminate quotations around love song and please describe the courtship song in a sentence or two for non-Drosophila readers

2) Line 74. I don't think it's correct to refer to the adult as hatching. I believe this is eclosion.

3) I understand the complexity in measuring particle velocity and thus why the authors reported dB in arbitrary units. Perhaps the authors though could make a statement at line 163 about the relationship between sound intensity and particle velocity. Are they linear? Does a double in sound intensity result in at least a near doubling in particle velocity. Some incite for readers who don't think about particle velocity might be beneficial.

4) Line 208. Remove about. The latency should be directly measured and not roughly estimated.

5) Line 209 - 210. The phrase in the parentheses feels like a sentence fragment. Please revise it.

6) Figure 4. Remove the figure title. Perhaps swap it for the y-axis and use no figure title.

7) Line 267. Remove "in passing". It's either been reported or not.

8) Line 289. Remove the whole sentence stating that a direct measurement of GF activity would be more advantageous. It's unnecessary and such direct measurements have their own limitations (equipment expenses, more difficult, invasive, etc).

9) Figure 5. There are a lot of non GFN spikes in these recordings that don't seem reliably time locked to stimulation. Perhaps mentioning that these are from other non-identified neurons should be mentioned in the figure legend.

10) For the diptheria toxin experiments, was there any independent validation that the toxin killed these neurons? Co-expression of GFP with the toxin is a reasonable what to do this.

11) The sentence in line 406 is a very long run on sentence and was difficult to understand. Simply chop it up into smaller more comprehend-able statements.

12) Line 439, eliminate "never". Just state, "because JON PSP's do not go above threshold in this configuration ...

13). Please revise the writing of point (5) starting on line 439 of the manuscript. It is a really long sentence and it causes confusion. I am not sure that somatic recordings are very useful for assuming spike thresholds as the spike initiation site might be very far from the soma. I simply did not understand point (5).

Reviewer #2: This is a very well written manuscript. I have only minor comments.

The manuscript describes how the sensitivity of auditory pathways in flies changes with age, but that their output connection to a pair of descending giant fibers does not. This is shown using an elegant 'trick' to measure the connection strength between the auditory neurons and the giant fibers: An additional electrical stimulus activates visual pathways that converge onto the same giant fibers but this input to the giant fibers is kept subthreshold. Only visual and auditory stimulus together 'sum up' and elicit action potentials in the giant fibers.

The experiments are well described and the conclusions are well formulated. The author has a very nice way of discussing alternative interpretations and laying out arguments for and against these interpretations.

Line 28: The sentence describing the 40% decrease is difficult to understand. Similarly, the following few sentences could be better. They make sense once one has read the whole paper, but as a first time reader they are difficult. The end of the introduction gives a nicer summary of the results, for example.

Line 44: "motor neuron" or "motoneuron". Also line 284.

Line 49: I think you can deleted the "somewhat uncoordinated"

Line 63: The shakeb should be designated as innexins and with a reference.

Line 84: I suggest to use "the strength of the connection between auditory input and GF"

Line 96: "Changes" in food probably does not mean different types of food? Please clarify in the text.

Line 164: Sound duration and frequency should be given. See also line 197: What does i.e. mean here? Were different frequencies used?

Line 166 and following: Maybe this could be improved a bit by explaining the general idea at the beginning of the paragraph: that an electrical stimulus activates optical pathways that are kept subthreshold, but enable the auditory stimulus to become suprathreshold and elicit GF activity. This would also help for later (line 384).

Line 350. 1ms delay to what?

Line 404 / 405: Maybe this sentence would be better if split into two.

Line 427: It's not clear to me why this is a combined results / discussion section. The discussion part clearly starts in line 427, so why not call it discussion here? (just a suggestion)

Line 441. This half sentence needs to be moved to line 433.

Figure 6: the averaged inset needs a scale bar and state the number of averaged responses.

Figure 6B: How do we know that the large spike is not the 'usual' GF spike, while the smaller spike before is just a subthreshold reflection of synaptic input? Maybe this could be stated clearly in the text.

6. PLOS authors have the option to publish the peer review history of their article (what does this mean?). If published, this will include your full peer review and any attached files.

Reviewer #1: No

Reviewer #2: Yes: Wolfgang Stein

---

## [Author Response · Author response to Decision Letter 0]

3 Dec 2019

PONE-D-19-27717 Response to Reviewers.

Line 28: Please, revise for clarity the sentence describing the 40% decrease and the following few sentences.

These sentences have been rewritten

Line 44: "motor neuron" or "motoneuron". Also line 284.

Corrected to “motor neuron”

Line 49: I think you can delete the "somewhat uncoordinated"

Deleted

Line 63: The shakeb should be designated as innexins and with a reference.

This has been done as requested

 Line 70. Eliminate quotations around love song and please describe the courtship song in a sentence or two for non-Drosophila readers

As requested

Line 84: Consider using "the strength of the connection between auditory input and GF"

As suggested

Line 96: "Changes" in food probably does not mean different types of food? Please clarify in the text.

Clarified to: “changed to fresh food vials weekly”

Please, explain at line 163 the relationship between sound intensity and particle velocity. Are they linear? Some incite for readers who is not familiar with particle velocity might be beneficial.

I had incorrectly used the term “sound intensity” rather loosely here – it has a more precise meaning in acoustics, to do with power per unit area. I have changed this to “sound level” throughout. As I understand it, in the near-field conditions used here, sound particle velocity is apparently not directly proportional to sound pressure, as it is at greater distances from the source. I have made this clear in the Methods.

Line 164: Sound duration and frequency should be given. See also line 197: What does i.e. mean here? Were different frequencies used?

After further research, it appears that use of the term frequency has little meaning when applied to a sound transient or impulse, so I removed any references to the stimulus frequency, only referring to its initial peak duration and amplitude.

Line 166 and following: Maybe this could be improved a bit by explaining the general idea at the beginning of the paragraph: that an electrical stimulus activates optical pathways that are kept subthreshold, but enable the auditory stimulus to become suprathreshold and elicit GF activity. This would also help for later (line 384).

As requested.

Line 208. Remove about. The latency should be directly measured and not roughly estimated.

Done

Line 209 - 210. The phrase in the parentheses feels like a sentence fragment. Please revise it.

Done

Line 267. Remove "in passing". It's either been reported or not.

Done

Line 289. Remove the whole sentence stating that a direct measurement of GF activity would be more advantageous. It's unnecessary and such direct measurements have their own limitations (equipment expenses, more difficult, invasive, etc).

Good point. Done

Line 350. 1ms delay to what?

This was expressed badly before. I was referring to the delay from the TTL pulse at the trace onset. This has been made clearer in the text.

Line 404 / 405: Maybe this sentence would be better if split into two.

Duly split.

Split sentence in line 406 into smaller statements.

Done

Line 427: It's not clear why this is a combined results / discussion section. The discussion part clearly starts in line 427, so why not call it discussion here? (just a suggestion)

You are absolutely right – I have done as suggested.

Line 441. This half sentence needs to be moved to line 433.

Done as suggested

Figure 4. Remove the figure title. Perhaps swap it for the y-axis and use no figure title.

Good suggestion

Figure 5. There are a lot of non GFN spikes in these recordings that don't seem reliably time locked to stimulation. Perhaps mentioning that these are from other non-identified neurons should be mentioned in the figure legend.

Mentioned.

Review Comments to the Author

Reviewer #1:

1) Line 70. Eliminate quotations around love song and please describe the courtship song in a sentence or two for non-Drosophila readers

Done

2) Line 74. I don't think it's correct to refer to the adult as hatching. I believe this is eclosion.

Absolutely correct! Fixed.

3) I understand the complexity in measuring particle velocity and thus why the authors reported dB in arbitrary units. Perhaps the authors though could make a statement at line 163 about the relationship between sound intensity and particle velocity. Are they linear? Does a double in sound intensity result in at least a near doubling in particle velocity. Some incite for readers who don't think about particle velocity might be beneficial.

I had used the term “sound intensity” rather loosely here – it has a more precise meaning in acoustics, to do with power per unit area. I have changed this to “sound level” throughout. Unfortunately, in the near-field conditions used here, sound particle velocity is apparently not directly proportional to sound pressure, as it is at greater distances from the source. I have made this clear in the Methods.

4) Line 208. Remove about. The latency should be directly measured and not roughly estimated.

As above, done

5) Line 209 - 210. The phrase in the parentheses feels like a sentence fragment. Please revise it.

As above, done

6) Figure 4. Remove the figure title. Perhaps swap it for the y-axis and use no figure title.

Done

7) Line 267. Remove "in passing". It's either been reported or not.

Done

8) Line 289. Remove the whole sentence stating that a direct measurement of GF activity would be more advantageous. It's unnecessary and such direct measurements have their own limitations (equipment expenses, more difficult, invasive, etc).

Done

9) Figure 5. There are a lot of non GFN spikes in these recordings that don't seem reliably time locked to stimulation. Perhaps mentioning that these are from other non-identified neurons should be mentioned in the figure legend.

Done

10) For the diptheria toxin experiments, was there any independent validation that the toxin killed these neurons? Co-expression of GFP with the toxin is a reasonable what to do this.

This would have been the most elegant way to show that the GF was completely missing and I was originally planning to do this. However the loss of many stocks after hurricane Maria prevented me from doing these particular experiments. I did however look at the neck connective from the diphtheria toxin expressing animals and was unable to see the GF axons with Nomarski optics, when they are normally very prominent. I have now mentioned this in the Results. It must be pointed out that it is not necessary that the GF be completely eliminated, just poisoned, in order for it to stop spiking. I have also altered the word “ablated” to “poisoned”.

11) The sentence in line 406 is a very long run on sentence and was difficult to understand. Simply chop it up into smaller more comprehend-able statements.

Done

12) Line 439, eliminate "never". Just state, "because JON PSP's do not go above threshold in this configuration ...

Done

13). Please revise the writing of point (5) starting on line 439 of the manuscript. It is a really long sentence and it causes confusion. I am not sure that somatic recordings are very useful for assuming spike thresholds as the spike initiation site might be very far from the soma. I simply did not understand point (5).

This part has been rewritten, and the threshold estimate moved to point 1. The point about the actual value of spike threshold is well taken, however, all these estimates are made from the point of view of the soma, since our only measurements of EPSPs come from patch recordings.

Reviewer #2: 

Line 28: The sentence describing the 40% decrease is difficult to understand. Similarly, the following few sentences could be better. They make sense once one has read the whole paper, but as a first time reader they are difficult. The end of the introduction gives a nicer summary of the results, for example.

This last part of the abstract has been rewritten to hopefully be clearer.

Line 44: "motor neuron" or "motoneuron". Also line 284.

Corrected

Line 49: I think you can deleted the "somewhat uncoordinated"

Deleted

Line 63: The shakeb should be designated as innexins and with a reference.

This has been done

Line 84: I suggest to use "the strength of the connection between auditory input and GF"

As suggested

Line 96: "Changes" in food probably does not mean different types of food? Please clarify in the text.

Clarified to mean fresh food vials weekly

Line 164: Sound duration and frequency should be given. See also line 197: What does i.e. mean here? Were different frequencies used?

After further research, it appears that use of the term frequency has little meaning when applied to a sound transient or impulse, so I removed any references to the stimulus frequency, only referring to its initial peak duration and amplitude.

Line 166 and following: Maybe this could be improved a bit by explaining the general idea at the beginning of the paragraph: that an electrical stimulus activates optical pathways that are kept subthreshold, but enable the auditory stimulus to become suprathreshold and elicit GF activity. This would also help for later (line 384).

Edited as requested

Line 350. 1ms delay to what?

This was expressed badly before. I was referring to the delay from the TTL pulse at the trace onset. This has been made clearer in the text.

Line 404 / 405: Maybe this sentence would be better if split into two.

Done

Line 427: It's not clear to me why this is a combined results / discussion section. The discussion part clearly starts in line 427, so why not call it discussion here? (just a suggestion)

They have been split into two sections, as suggested.

Line 441. This half sentence needs to be moved to line 433.

Edited

Figure 6: the averaged inset needs a scale bar and state the number of averaged responses.

Added to the figure and legend.

Figure 6B: How do we know that the large spike is not the 'usual' GF spike, while the smaller spike before is just a subthreshold reflection of synaptic input? Maybe this could be stated clearly in the text.

The small subthreshold response to eye stimulation could indeed also be from the visual circuit’s synaptic inputs to the GF, although I suspect that these are likely to be too small. I have tried to clarify this in the text.

---

## [Decision Letter · Decision Letter 1]

16 Dec 2019

A new method of recording from the giant fiber of Drosophila melanogaster shows that the strength of its auditory inputs remains constant with age

PONE-D-19-27717R1

Dear Dr. Blagburn,

We are pleased to inform you that your manuscript has been judged scientifically suitable for publication and will be formally accepted for publication once it complies with all outstanding technical requirements.

With kind regards,

Gennady Cymbalyuk, Ph.D.

Academic Editor

PLOS ONE

Additional Editor Comments (optional):

Reviewers' comments:

Reviewer's Responses to Questions

**Comments to the Author**

1. If the authors have adequately addressed your comments raised in a previous round of review and you feel that this manuscript is now acceptable for publication, you may indicate that here to bypass the “Comments to the Author” section, enter your conflict of interest statement in the “Confidential to Editor” section, and submit your "Accept" recommendation.

Reviewer #1: All comments have been addressed

Reviewer #2: All comments have been addressed

2. Is the manuscript technically sound, and do the data support the conclusions?

Reviewer #1: Yes

Reviewer #2: (No Response)

3. Has the statistical analysis been performed appropriately and rigorously? 

Reviewer #1: Yes

Reviewer #2: (No Response)

4. Have the authors made all data underlying the findings in their manuscript fully available?

Reviewer #1: Yes

Reviewer #2: (No Response)

5. Is the manuscript presented in an intelligible fashion and written in standard English?

Reviewer #1: Yes

Reviewer #2: (No Response)

6. Review Comments to the Author

Reviewer #1: The Author has adequately addressed the issues that I raised in the previous version of the manuscript. As a result, the current version is now clearer and better supported.

Reviewer #2: (No Response)

7. PLOS authors have the option to publish the peer review history of their article (what does this mean?). If published, this will include your full peer review and any attached files.

Reviewer #1: No

Reviewer #2: Yes: Wolfgang Stein

---

## [Editor Report · Acceptance letter]

17 Dec 2019

PONE-D-19-27717R1 

A new method of recording from the giant fiber of *Drosophila melanogaster* shows that the strength of its auditory inputs remains constant with age 

Dear Dr. Blagburn:

I am pleased to inform you that your manuscript has been deemed suitable for publication in PLOS ONE. Congratulations! Your manuscript is now with our production department. 

With kind regards,

on behalf of

Dr. Gennady Cymbalyuk 

Academic Editor

PLOS ONE